# Sarcopenia as a Mediator of the Effect of a Gerontogymnastics Program on Cardiorespiratory Fitness of Overweight and Obese Older Women: A Randomized Controlled Trial

**DOI:** 10.3390/ijerph17197064

**Published:** 2020-09-27

**Authors:** Pablo Jorge Marcos-Pardo, Noelia González-Gálvez, Gemma María Gea-García, Abraham López-Vivancos, Alejandro Espeso-García, Rodrigo Gomes de Souza Vale

**Affiliations:** 1Research Group on Health, Physical Activity, Fitness and Motor Behaviour (GISAFFCOM), Physical Activity and Sport Sciences Department, Faculty of Sport, Catholic University San Antonio of Murcia, 30107 Murcia, Spain; pmarcos@ucam.edu (P.J.M.-P.); gmgea@ucam.edu (G.M.G.-G.); alvivancos@ucam.edu (A.L.-V.); aespeso@ucam.edu (A.E.-G.); 2Active Aging, Exercise and Health/HEALTHY-AGE Network, Consejo Superior de Deportes (CSD), Ministry of Culture and Sport of Spain, 28040 Madrid, Spain; rodrigovale@globo.com; 3Exercise Physiology Laboratory, Estacio de Sa University, 20261-063 Rio de Janeiro, Brazil

**Keywords:** Sarcopenia, gait speed, cardiorespiratory responses, walking, physical fitness, older people

## Abstract

The objectives were to analyze the effect of a gerontogymnastics program on functional ability and fitness on overweight and obese older woman and to understand if sarcopenia mediates its effect. This randomized controlled trial involved 216 overweight and obese women. The experimental group (EG) carried out 12 weeks of a gerontogymnastics program. The assessment was of gait speed, cardiorespiratory fitness, functional capacity, and muscle strength. EG showed significant improvements in almost every test. When the effect of training was adjusted by gait speed, the improvement of the 6 min walk test (MWT) for the trained group was no longer significant (*p* = 0.127). The improvement of the 6 MWT was significantly and positively associated with the 10 m test (*β* = −10.087). After including the 10-m test in the equations, the association between the 6MWT and carrying out the training program decreased but remained significant (β = −19.904). The mediation analysis showed a significant, direct and indirect effect with a significant Sobel test value (*z* = 6.606 ± 7.733; *p* = 0.000). These results indicate that a gerontogymnastics program improves functional capacity and fitness; and the effect of a gerontogymnastics program on CRF is mediated by sarcopenia in older women who are overweight and obese.

## 1. Introduction

Sarcopenia is a progressive disease that involves the loss of muscle mass and strength [1]. It is associated with aging and causes a decrease in functional capacity, increasing the risk of falls and negatively affecting the quality of life, and in many cases may require hospitalization or rehabilitation [2,3]. Sarcopenia affects about 5 to 13% of individuals in their 60s and 70s, and 11 to 50% of octogenarians [4]. From the age of 40, there is a decline of muscle mass of approximately 8% per decade, which around the age of 70 can reach up to 15% per decade [5]. Besides, older women are more susceptible to present sarcopenia, as opposed to young women and men [6].

Age is associated with loss of mass, strength, and muscle power [7,8,9]. The loss of muscle mass and strength increases the risk of falls and potential fractures and contributes to the loss of functionality, impedes the older individual’s ability to live independently [10], and results in a worse quality of life of a person [11,12,13]. The ability of the leg muscles to produce strength is a key factor in maintaining the older person’s balance and walking speed [14]. According to the definition by the European Working Group on Sarcopenia in Older People (EWGSOP) [3,15], the diagnosis of sarcopenia can be made by assessing the low muscle mass, plus low muscle strength or low physical performance. The most commonly used parameters to measure muscle mass loss are dual energy x-ray absorptiometry (DEXA) and bioelectrical impedance analysis (BIA), to measure muscle strength it’s handgrip strength, and to measure physical performance it’s short physical performance (SPPB) and gait speed (GS) (Table 1).

The EWGSOP suggests the gait speed test as an easy and valid method for assessing physical performance [3,15,16]. Gait speed has been performed to evaluate various health-related factors such as physical functions, health status [17,18,19], and quality of life [20]. A well-known meta-analysis of 2888 older people set the minimum health threshold for gait speed at ≥ 0.8 m/s [21]. The promotion of physical exercise programs can prevent or even reverse the loss of functional capacity associated with sarcopenia if they emphasize improving the gait speed of the older population [22].

The increased fat mass associated with sarcopenia has been linked with a higher incidence of chronic diseases in older people [5,23,24], and developing sarcopenia contributes to the development of cardiovascular and metabolic diseases [25]. Being sedentary or not very physically active contributes to sarcopenia, as shown by research result, which have also linked age to the development of sarcopenia. The ICD-10 code for sarcopenia in 2016 was established by the World Health Organization to promote effective therapeutic strategies that include physical exercise to prevent loss of muscle mass and function with aging [26].

Recent research suggests effective intervention strategies to combat sarcopenia that include physical exercise, and more specifically, strength training [27,28,29,30,31]. Also, maximum oxygen consumption (VO^2^max) is a measurement of cardiorespiratory fitness (CRF), which can predict longevity in older adults [32]. People who have a higher risk of cardiovascular disease (CVD) and mortality are those who have a lower CRF [33]. For older individuals, regular aerobic exercise helps them to attain better VO^2^ values [34]. With age, exercise that includes long-term aerobic exercise can help combat the effects of sarcopenia [35].

Being overweight and obese, coupled with a poor physical condition, are related to aging and are also associated with the risk of death from chronic diseases. Therefore, strategies are needed to encourage changes in body composition and physical condition [36,37]. Exercise programs, such as gerontogymnastics, which include resistance and aerobic training, are an optimal strategy for maintaining muscle mass and its protective effects against a variety of chronic diseases [38,39,40,41,42]. However, older adults with low functional capacity may not be able to develop resistance programs leading to improved CRF, due to their low fitness [43]. Besides, the improvement of CRF could also be influenced by improved strength [44].

Therefore, the objectives of the study were the following: a) to analyze the effect of a gerontogymnastics program with overweight and obese older women (≥65 years old) on functional ability and fitness, and b) to understand if sarcopenia mediates the effect of a gerontogymnastics program on cardiovascular fitness. We hypothesized that the older women who participate in the trained group will show improvements in all the tests, whereas the control group will not show changes, and that sarcopenia will mediate the effect of the program on cardiorespiratory capacity.

## 2. Materials and Methods

### 2.1. Design

The study was conducted from September to December 2019, with a total of 12 weeks of training. This is a randomized controlled trial study that investigates the effect of gerontogymnastics program with overweight and obese older women on functional ability and physical fitness. This study trial followed the Consolidated Standards of Reporting Trials (CONSORT) guidelines. Older women were informed of the study and signed an informed consent to participate, according to the process approved by the local ethics committee (CE111908) of the San Antonio Catholic University of Murcia (Spain) and in accordance with the Declaration of Helsinki. This study was conducted at a women’s association and sport science laboratory from Region de Murcia (Spain).

### 2.2. Participants

Participants were recruited through advertisements in women’s associations, senior centers, and presentations in the local community. Recruitment was done before registration for convenience and accessibility to the sample, but the intervention did not begin until the registration was completed. Inclusion criteria were: (a) at least 1 year not engaged in a structured exercise program, (b) having a body mass index between 25 and 29.9 (overweight) or between 30 and 34.9 (obese), (c) women aged between 65 and 90 years old, and (d) being physically independent. The exclusion criteria were: (a) having musculoskeletal injuries or limitations that could affect the person’s health and physical performance; (b) being under a doctor’s prescription for taking medication that could influence physical performance; (c) no regular attendance at the proposed sessions.

Sample size and power were established in connection with the 10-m walk test in a previous study [45]. An estimated error of 0.045 s and a significance level of *α* = 0.05 were utilized. A valid sample size for a confidence interval of 95% was 207.84. Based on previous research, a dropout rate of 10% was assumed; therefore, 230 participants were recruited.

Participants were divided to a trained group (TG) and a control group (CG). Thus, the experimental sample of the study consisted of 230 women, being electronically randomized [46] into the TG (115 subjects) and the CG (115 subjects). A researcher who was not involved in participant recruitment performed the randomization.

Two-hundred-and-sixteen older women aged between 65 and 88 years old volunteered (mean ± standard deviation [SD]; age = 68.26 ± 4.19 years, body mass = 71.11 ± 10.66 kg, height = 1.55 ± 0.07 m; BMI = 29.97 ± 3.86) and completed the study (TG = 114; CG = 102). The CONSORT 2010 flow diagram is shown in Figure 1.

Anthropometric measurements were recorded. Weight (kg) was evaluated in light clothing without footwear to the nearest 0.1 kg by using an electronic scale, and height (cm) was measured using a stadiometer to the nearest millimeter (Seca 763 digital scale, Birmingham, UK). Body mass index (BMI) was calculated by dividing their weight in kilograms by their height in square meters (kg/m^2^). All anthropometric measurements were completed by experienced and well-trained persons (ISAK level 1 and 2 certificate). The same researchers performed all the measurements in a single session between the hours of 9:00 and 13:00 without warming up and allowing for a 5-min break between tests.

### 2.3. Procedure

#### 2.3.1. Trained Group

The gerontogymnastics program was implemented following the recommendations of the Otago Exercise Program (OEP) [47], as it is a renowned exercise program with widespread use at the international level that aims to improve strength and mobility to help in the prevention of falls. The gerontogymnastics program was planned principally to help prevent the risk of falls with exercise training that improves muscle strength, power, and balance in the lower extremities and cardiorespiratory endurance. A professional Sports Physical Educator directed the training. The total training group was divided into subgroups of a maximum of 20 subjects for security and the correct direction by the trainer. The participants in the TG trained for 1 h, three times a week for the 12 weeks of intervention (36 sessions). The training session consisted of: (a) 10 min of warm-up, consisting of joint mobility and stretching of the main muscle groups involved; (b) 30 min of an exercise circuit (three sets of 15 repetitions in each exercise, with 2 min rest between sets). The exercise circuit consisted of 12 exercises, of which, seven focused on strength: knee extension, squat, knee curl, leg press, elbow curl, chest press, and shoulder overhead press using the OMNI resistance exercise scale [48]; five focused on balance: walking on marked lines on the floor, walking on tiptoes, walking sideways, walking on heels, and walking from heel to toe; and (c) 20 min of cardiovascular exercises. The cardiovascular exercise consisted of walking at maximum speed without running to maintain a moderate to hard level of perception of exertion [49]. The training sessions were held on non-consecutive days to facilitate recovery.

#### 2.3.2. Control Group

The CG participants were asked to carry out their normal life and not to alter their habits during the study period, and they did not practice any physical activity or exercise program.

#### 2.3.3. Assessment of gait speed

Gait speed was assessed with a 10-m test. The time in seconds that the person took to walk 10 m was analyzed, with the person walking at their usual pace. This test has been widely used in large epidemiological studies, showing high concurrent and predictive validity [45,50,51,52,53]. The results of previous studies [45] indicate an excellent relationship between the 4-m test and the 10-m test (ICC = 0.959 and 0.976, respectively), with a good average between both tests (ICC = 0.867). The 10-m test proved to be somewhat better. This test is a valid method for predicting sarcopenia [54]. The reference value for gait speed in the 10-m test is 0.8 m/s [55]. In the present study, in order to obtain reliable measurements, two photocells were placed at the beginning and the end of a 10-m lane, and through a connection to a computer, recorded the time spent in carrying out the test (MuscleLab, Ergotest, Langesund, Norway). The older women were asked to stand at the starting line mark and walk at their usual pace at the sound signal. Two attempts were made and the average value between the two repetitions was recorded.

#### 2.3.4. Assessment of Cardiorespiratory Fitness Level

Aerobic endurance was assessed using the 6-min walk test (6 MWT). The 6 MWT has been shown to be a valid, reliable, objective, inexpensive, and easy test used to evaluate cardiorespiratory capacity [56,57,58,59,60]. It is a simple test to perform and is better related to the person’s daily life activities than other tests [58,59]. It is used to measure an individual’s sub-maximum aerobic capacity while walking for 6 min.

It is suggested that this test should be performed on a flat surface that allows walking for 20 to 30 m. The subject should be relaxed and wear comfortable clothing and shoes and the heart rate of each subject was recorded with a POLAR 400 heart rate monitor just before the start of the test and just after the end. The route was marked every 5 m and cones were placed at the turning area. The subject was walking at a pace appropriate to his/her condition, being able to stop or slow down if he/she is fatigued and resume as soon as possible. The trainer can motivate the subjects with phrases such as “You are doing well”, and the total meters walked is recorded [61]. This test has good reliability (ranging from 0.95 to 0.97) [62].

#### 2.3.5. Assessment of Functional Capacity

The Latin American Group for Maturity (GDLAM) protocol is used to evaluate the functional capacity in older adults [27,63,64]. The battery consists of five tests: walking 10 m; rising from a sitting position; standing up from a prone position on the floor; getting up from a chair and moving around; and the putting on and taking off a T-shirt test. These tests were to calculate the GDLAM functionality index (GI) using a mathematical formula. The material needed for carrying out the tests consisted of a standard chair with a height of 48 cm from the seat to the floor, a digital chronometer, four cones, a sports mat, and a metal measuring tape. The magnitude of the statistical significance demonstrated high reliability (*r* = 0.9; *p* < 0.001) and validity [63].

#### 2.3.6. Assessment of Muscle Strength

Two tests from the “Senior Fitness Tests” (SFT) battery [59,65] were used to assess strength variables: extension flexion elbow test and lift chair 30 s test. The extension flexion elbow test measures the muscle strength of the upper extremity. The subject, while sitting on a chair, was asked to perform the maximum number of repetitions for 30 s with a dumbbell (2.3 kg for women). The lift chair 30 s test reflects lower body strength. The participant was asked to sit on a chair with his arms across his chest and perform the most sitting and standing repetitions for 30 s. Reliability and validity indicators for the standards ranged between 0.79 and 0.97 [66].

### 2.4. Data Analysis

The normality of the data was evaluated using the Kolmogorov–Smirnov test, and Mauchly’s W-test was used to analyze the normality and the sphericity of the data. The inter- and intra-groups differences and the interaction between groups and time were analyzed with a two-way ANOVA with repeated measurements of one factor (time). Also, an ANCOVA (adjusted for gait speed) with repeated measurement of one factor (time) was used. To check intra-groups change, the post-hoc Bonferroni test and the Wilcoxon signed-rank test were used to evaluate the statistical significance of parametric and non-parametric variables, respectively. The Mann–Whitney test was used to check for inter-group differences for non-parametric variables. The partial eta-squared (η2p) for variance analysis was used to calculate the size effect, and this was defined as small: ES ≥ 0.10; moderate: ES ≥ 0.30, large: ES ≥ 1.2; or very large: ES ≥ 2.0, with an error of *p* ≤ 0.05 utilized [67].

To determine if the effect on the 6MWT test was mediated by the change in the 10-m test, the analysis of the mediation variables was performed using the Process macro for SPSS (SPSS Inc, Chicago, Illinois). A resample procedure of 10,000 bootstrap samples for non-parametric variables was utilized, [68] and the classical Baron and Kenny step regression method was used for parametric ones. [69]. In order to analyze the statistical significance of the mediation effect, the Sobel test was used [70]. If after the mediation, the independent variable was no longer associated with the dependent variable, it was considered complete mediation. However, if after the mediation, the independent variable was reduced but was still significant, it was considered partial mediation. The statistical analysis was performed using IBM SPSS Statistics (version 24.0), and an error of *p* ≤ 0.05 was set for the analysis.

## 3. Results

The characteristics of the participants are shown in Table 2. The TG showed significant improvements in the 10-m test (*p* < 0.000), the 6 MWT (*p* = 0.001), stand from siting test (*p* < 0.000), the rising from sitting test (*p* < 0.000), the rise from the floor test (*p* < 0.000), the t-shirt test (*p* < 0.000), the GDLAM index (*p* < 0.000), the extension and flexion elbow test (*p* < 0.000), and the lift chair 30 s test (*p* < 0.000). TG did not show changes in the stand-up and go test (*p* = 0.150) and showed an increase of BMI (*p* = 0.021) but with a very low effect size (ES = 0.03).

The CG experienced a significant decrease in the 10-m test (*p* < 0.000), 6 MWT (*p* = 0.011), rise from the floor test (*p* = 0.032), stand-up and go test (*p* < 0.000), and extension flexion elbow test (*p* < 0.000), and showed a significant improvement in the rise from the floor test (*p* = 0.032), although they did not show changes in the rest of the tests. Although both groups showed a significant improvement in the rise from the floor test, the effect size was small for the CG (ES = 0.12), whereas the effect size for the TG was large (ES = 0.72) (Table 3).

Table 4 shows the differences between groups in the changes pre- and post-test. The results show a difference between groups for all the functional and fitness tests in favor of TG.

When the effect of training was adjusted according to gait speed, the improvement of the 6 MWT for TG was no longer significant (TG = difference post-pre (M ± SD): −9.476 ± 6.178; *p* = 0.127; CI 95% (Mpost–Mpre): −21.653;2.702; CG = difference post-pre (M ± SD): 5.601 ± 6.633; *p* = 0.399; CI 95% (Mpost–Mpre): −7.473;18.675).

The improvements in the 6 MWT (*β* = −32.129) and 10-m test (*β* = 1.689) were significantly associated with carrying out the training program (TG). The improvement in the 6 MWT was significantly and positively associated with the 10 m test (*β* = −10.087). After including the 10 m test in the equations, the association between the 6MWT and carrying out the training program (TG) decreased, although it remained significant (*β* = −19.904). The mediation analysis showed significant, direct and indirect effects with a significant Sobel test value (*z* = 6.606 ± 7.733; *p* ˂ 0.000). These results indicate that gait speed (10 m test) acts as a mediator on the effect of the exercise program on the 6 MWT (Figure 2).

## 4. Discussion

The first objective of this randomized controlled trial was to analyze the effect of a gerontogymnastics program for overweight and obese older women on functional ability and fitness. Significant improvements in functional capacity (10-m test, rise from sitting test, rise from the floor test, t-shirt test, and GDLAM index), CRF (6MWT) and muscle strength and endurance (extension and flexion elbow and lift chair 30 s test) were reported by the group that carried out the intervention program. The CG showed a significant decrease in the 10-m test, 6MWT, rise from the floor test, stand-up and go test, and extension flexion elbow test; and did not show changes in the rest of the tests. In connection with the stand-up and go test, the TG did not show any changes; however, the CG experiment showed a significant decrease. This could be interpreted as the intervention program preventing the physical decline due to age. Although both groups showed a significant improvement in the rise from the floor test, the effect size was small for the CG (ES = 0.12), whereas the effect size for the TG was large (ES = 0.72); and there was also an inter-groups difference that indicated that the TG significantly improved more than the CG.

Other studies that implemented a similar exercise program also reported improvements in functional capacity and fitness [42,47,71,72,73]. These studies implemented their programs from 8 to 18 weeks, with a frequency of three times per week and a session duration ranging from 50 min to 60 min. Related to this, our study included different sets of exercises for strength training, balance, and cardiovascular endurance. This exercise program is adapted to older women who are overweight and obese.

A 12-week, low-to-moderate-intensity at maximal fat oxidation intensity (FATmax; 37–54% VO_2_max) exercise program for overweight and obese older women resulted in favorable changes in body composition and functional capacity in the exercise (training) group, compared with the outcomes of the control group [74]. Another study revealed that 12 weeks of elastic resistance training exerted positive effects on functional mobility outcomes of older women with sarcopenic obesity [75]. No prevalence of obesity, a higher level of physical activity, and baseline grip strength were associated with better mobility performance among the older population [76]. Physical activity mitigated the deleterious effects of the loss of functional capacity and muscle strength in obese individuals, highlighting its importance in the creation of strategies for the preservation of physical function with age [77]. These results support the evidence that a 12-week gerontogymnastics program that included endurance and strength training exercises improves functional capacity, CRF, and strength and endurance of musculature of overweight and obese older women; and could thus delay the harmful effects of aging.

The second objective of this study was to understand if sarcopenia mediated the effect of a gerontogymnastics program on cardiovascular fitness. The major finding of our study was that an improvement in CRF was associated with an improvement in gait speed, in consonance with the decrease in sarcopenia. Our results are in agreement with a previous study, showing a connection between CRF and gait speed and sarcopenia [78]. In our study, sarcopenia acted as a partial mediator on the association between carrying out a gerontogymnastics program and improved CRF. To the best of our knowledge, this is the first randomized controlled trial with an analysis of the mediation that assesses how sarcopenia influences the effect of an exercise program on CRF.

A recent study [79] assessed 527 women aged 75 years and older (79.7 ± 3.5) in a cross-sectional study. The objective of this study was to investigate if the connection between physical activity and gait speed was mediated by strength and weight. These authors reported that the association between physical activity and gait speed was partially mediated by the absolute and relative strength of the lower limbs and that muscle mass partially mediated the relationship between physical activity and muscle strength.

On the other hand, it has been demonstrated that there is a connection between walking balance and strength [80] and that sarcopenia influences walking balance [81]. A study with older adults with mild to moderate frailty improved their CRF but at a modest level [82]. This suggests that it will be necessary to increase leg strength to further increase walking speed, in order to improve CRF. In this sense, a study was performed to determine the mechanisms responsible for the effect of exercise training on CRF in older adults, utilizing a strength training program before an endurance training program, with a sample of 22 older adults, to improve their functional capacity [44].

In agreement with another study [43], older adults with declined functionality were not able to participate in endurance training until they improved their neuromuscular capacity. Therefore, endurance training for older women should be performed with previous strength and resistance training to achieve the highest CRF adaptations.

It has also been reported that gait speed, muscle mass, and sarcopenia are strongly associated with functional capacity [17,18,19]. However, our study expands this finding by showing that sarcopenia is not just a predictor, but also an important mediator of the effect of an exercise program on another important factor for the health such as CRF.

Strong research methodologies, such as a randomized clinical trial with a blinded examiner, is one of the strengths of the present study. Also, to minimize the risk of bias, a large sample size was utilized. However, our study is not without limits. This research was developed with older women who were overweight and obese, and thus, we are not able to generalize the result to other populations of interest.

## 5. Conclusions

A gerontogymnastics program improves the functional capacity and fitness of older women who are overweight and obese. Sarcopenia acts as a mediator of the effect of a gerontogymnastics program on CRF in overweight and obese older women.

In this sense, the results support the new interest in changing the type of intervention and could be used to suggest that the improvements in strength, gait speed, and reduction of sarcopenia at the start of the exercise program could be needed to secure or improve the effects of the program on CRF and help improve the health of overweight and obese older people.

## Figures and Tables

**Figure 1 ijerph-17-07064-f001:**
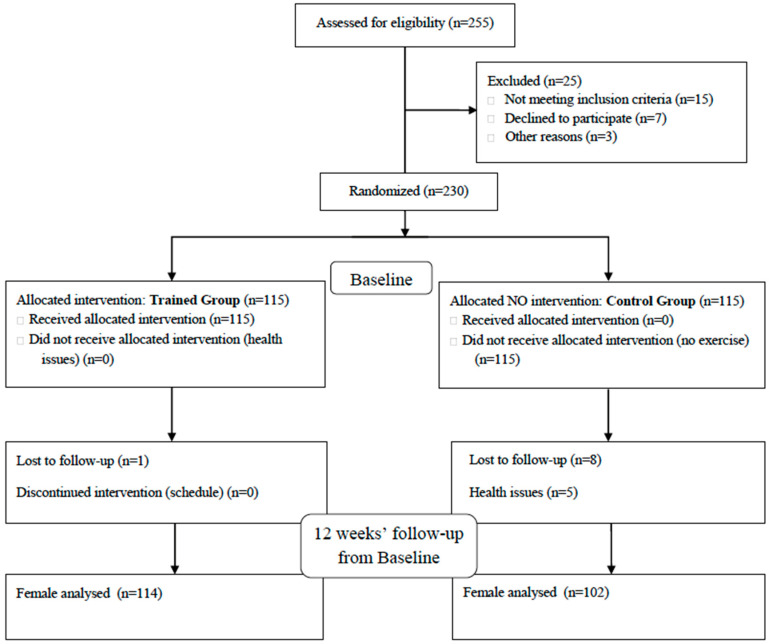
Flow diagram of the sample.

**Figure 2 ijerph-17-07064-f002:**
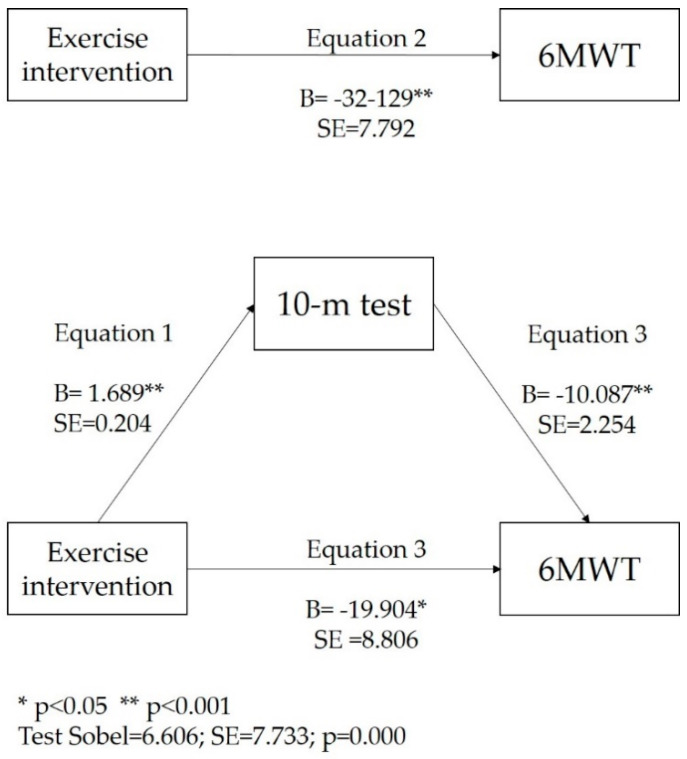
Mediation of exercise intervention and 6MWT by 10-metres test. ** *p* ˂ 0.001; * ˂ 0.05.

**Table 1 ijerph-17-07064-t001:** Sarcopenia: measurable variables and cut-off points [15].

Criterion	Measurement Method	Cut-off Points by Gender
Muscle mass	DEXA	Skeletal muscle mass index (SMI) (Appendicular skeletal muscle mass/height2) Men: 7 kg/m^2^ Women: 5.5 kg/m^2^
	BIA	SMI using BIA predicted skeletal muscle mass (SM) equation (SM/height^2^) Men: 8.87 kg/m^2^ Women: 6.42 kg/m^2^
Muscle strength	Handgrip strength	Men: <27 kg Women: <16 kg
	Chair stand	>15 s for five rises
Physical performance	SPPB SPPB score is a summation of scores on three tests: Balance, Gait Speed and Chair Stand. Each test is weighted equally with scores between 0 and 4—quartiles generated from Established Populations for Epidemiologic Studies of the Older people (EPESE) data (*n* = 6534). The maximum score on the SPPB is 12	SPPB ≤ 8 SPPB 0–6 Low performance SPPB 7–9 Intermediate performance SPPB 10–12 High Performance
	GS	GS ≤ 0.8 m/s

**Table 2 ijerph-17-07064-t002:** Characteristics of the participants.

	M ± SD
Age	68.03 ± 4.03
Weight (kg)	69.59 ± 11.28
Height (m)	1.55 ± 0.07
BMI (kg/m^2^)	29.97 ± 3.86
10-m test (s)	7.29 ± 1.51
6MWT (m)	460.29 ± 78.02
Stand from siting (s)	13.13 ± 3.31
Rise from the floor (s)	8.47 ± 3.64
Stand and go (s)	42.83 ± 5.03
T-shirt (s)	15.92 ± 6.16
GDLAM index	33.11 ± 5.18
Ex Flex Elbow 30 s (rep)	14.54 ± 4.16
Lift chair 30 s (rep)	10.98 ± 2.78

Legend: s = seconds; m = meters; rep = repetitions; M = Mean; SD = Standard Deviation; 6 MWT = 6 min walk test.

**Table 3 ijerph-17-07064-t003:** Differences pre- to post-test (intra-groups) for functional and fitness test.

		Pre-Test (M ± SD)	Post-Test (M±SD)	Difference Post-Pre (M±SD)	*p*	CI 95% (Mpost–Mpre)	ES
10-m test (s)	TG	6.93 ± 1.14	6.05 ± 0.94	0.878 ± 0.14	0.000	0.601; 1.155	0.77
	CG	7.59 ± 1.70	8.40 ± 1.68	−0.811 ± 0.15	0.000	−1.103; −0.518	0.47
6MWT (m)	TG	473.54 ± 70.42	491.06 ± 74.00	−17.528 ± 5.36	0.001	−28.083; −6.973	0.25
	CG	448.30 ± 79.33	433.70 ± 79.49	14.601 ± 5.66	0.011	3.442; 25.760	0.18
Stand from siting (s)	TG	13.56 ± 3.15	11.29 ± 2.69	2.272 ± 0.37	0.000	1.535; 3.010	0.72
	CG	12.96 ± 3.62	13.08 ± 3.70	−0.123 ± 0.4	0.756	−0.903; 0.657	0.03
Rise from the floor (s)	TG	8.92 ± 3.80	6.18 ± 3.27	2.740 ± 0.2	0.000	2.357; 3.124	0.72
	CG	8.44 ± 3.71	8.00 ± 3.92	0.444 ± 0.21	0.032	0.039; 0.850	0.12
Stand and go (s)	TG	41.07 ± 3.57	40.51 ± 4.04	0.566 ± 0.39	0.150	−0.206; 1.338	0.16
	CG	44.41 ± 5.44	46.36 ± 5.02	−1.950 ± 0.41	0.000	−2.766; −1.134	0.36
T-shirt (s)	TG	16.46 ± 6.94	11.73 ± 4.41	4.721 ± 0.37	0.000	3.999; 5.444	0.68
	CG	15.85 ± 5.66	15.36 ± 4.81	0.489 ± 0.39	0.208	−0.275; 1.252	0.09
GDLAM index	TG	33.20 ± 5.92	27.75 ± 4.50	5.448 ± 0.34	0.000	4.777; 6.118	0.91
	CG	33.52 ± 4.68	34.01 ± 3.83	−0.489 ± 0.36	0.176	−1.197; 0.220	0.10
Ex Flex Elbow 30 s (rep)	TG	14.59 ± 4.25	17.46 ± 4.55	−2.868 ± 0.28	0.000	−3.417; −2.320	0.67
	CG	14.63 ± 4.09	13.34 ± 3.64	1.284 ± 0.29	0.000	0.704; 1.865	0.31
Lift chair 30 s (rep)	TG	10.81 ± 2.46	13.09 ± 2.70	−2.281 ± 0.21	0.000	−2.699; −1.863	0.92
	CG	11.26 ± 3.17	11.15 ± 3.07	0.118 ± 0.22	0.600	−0.324; 0.560	0.04
BMI (kg/m^2^)	TG	31.13 ± 4.15	31.26 ± 4.13	0.13 ± 0.54	0.021	0.020; 0.242	0.03
	CG	28.68 ± 3.04	28.59 ± 3.17	−0.09 ± 0.66	0.125	−0.209; 0.026	0.02

Legend: TG = trained group; CG = control group; M = Mean; SD = Standard Deviation; ES = Effect Size; s = seconds; m = meters; rep = repetitions.

**Table 4 ijerph-17-07064-t004:** Differences pre to post-test (intergroups) for functional and fitness test.

	Group	Difference Post-Pre (M ± SD)	*F*	*p*	ES
10-m test (s)	TG	0.878 ± 0.14	68.220	0.000	0.242
	CG	−0.811 ± 0.15			
6MWT (m)	TG	−17.528 ± 5.36	17.000	0.000	0.074
	CG	14.601 ± 5.66			
Stand from siting (s)	TG	2.272 ± 0.37	19.354	0.000	0.083
	CG	−0.123 ± 0.4			
Rise from the floor (s)	TG	2.740 ± 0.2	65.676	0.000	0.235
	CG	0.444 ± 0.21			
Stand and go (s)	TG	0.566 ± 0.39	19.489	0.000	0.083
	CG	−1.950 ± 0.41			
T-shirt (s)	TG	4.721 ± 0.37	63.004	0.000	0.227
	CG	0.489 ± 0.39			
GDLAM index	TG	5.448 ± 0.34	143.774	0.000	0.402
	CG	−0.489 ± 0.36			
Ex Flex Elbow 30 s (rep)	TG	−2.868 ± 0.28	105.018	0.000	0.329
	CG	1.284 ± 0.29			
Lift chair 30 s (rep)	TG	−2.281 ± 0.21	60.373	0.000	0.220
	CG	0.118 ± 0.22			

Legend: s = seconds; m = meters; rep = repetitions; M = Mean; SD = Standard Deviation; 6 MWT = 6 min walk test; ES = effect size.

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
