# Peer review of "Sarcopenia as a Mediator of the Effect of a Gerontogymnastics Program on Cardiorespiratory Fitness of Overweight and Obese Older Women: A Randomized Controlled Trial"

_ijerph, 2020, doi:10.3390/ijerph17197064_

Round 1

Reviewer 1 Report

The manuscript from Marcos-Pardo and coworkers aimed to evaluate sarcopenia as a mediator of the effect of a gerontogymnastics program on cardiorespiratory fitness of overweight and obese older women. Applying a randomized controlled trial were randomized to training intervention or control.

This study adds knowledges about how exercise improve the quality of life in overweight and obese adult woman. However, several points needed to be clarified or improved.

Introduction
In page 1, line 14, the word t0o is incorrect.

In page 2, line 45. What are the parameter to define low muscle mass, low muscle strength or low physical performance for the sarcopenia diagnosis? which cutt-point are the most used?.

Desing
The study period is missing.

Participant
In page 4, line 119. The anthropometer used does not measure height.

In page 4, line 121. The researchers who performed all the measurements are nutritionist, doctors, or what kind of experience have with the anthropometric measurements?.

The numer of appointments are missing.

In page 4, line 120. "Its physical fitness and functional capacity was recorded". I recommend explaining better or detailing this information.

Trained group
Some information are missing. The session were supervised? the participants training by groups, or alone? if they trained alone, how was the control/record of the exercise program?.

Results

Could the authors have more information about the fat mass, lean mass or muscle mass of the participants?.

In anthropometric results, the authors should be more informative about the changes at the end of the intervention.

If the gait speed test could be used to diagnosis of sarcopenia. The author estimated the prevalence in this population? at the end of the intervention how many people improve?

In page 7, line 234. Homogenize EG / TG

Discussion
In page 8, line 268. A 12 weeks, "low to moderate-intensity exercise program". What intensity was applied?

Author Response

REVIEWER 1

Comments and Suggestions for Authors

The manuscript from Marcos-Pardo and coworkers aimed to evaluate sarcopenia as a mediator of the effect of a gerontogymnastics program on cardiorespiratory fitness of overweight and obese older women. Applying a randomized controlled trial were randomized to training intervention or control.

This study adds knowledges about how exercise improve the quality of life in overweight and obese adult woman. However, several points needed to be clarified or improved.

Introduction

In page 1, line 42, the word t0o is incorrect. 

  • The error in the writing has been corrected. Thank you very much for the suggestion

In page 2, line 45. What are the parameter to define low muscle mass, low muscle strength or low physical performance for the sarcopenia diagnosis? which cutt-point are the most used?.

  • The parameters and cut-off points have been included in the text for these variables. Thanks for the suggestion. It is more complete for the reader now.

Design

The study period is missing.

  • The date of the study has been incorporated. Thank you very much for the suggestion

Participant

In page 4, line 119. The anthropometer used does not measure height.

  • This error has been corrected and we have placed the information of the measuring device used. Thank you very much for the correction

In page 4, line 121. The researchers who performed all the measurements are nutritionist, doctors, or what kind of experience have with the anthropometric measurements?.

  • The researchers who performed all the measurements are Doctors of Sports Sciences with ISAK level 1 and 2 for anthropometric measurements.

The number of appointments are missing.

  • Number of total session has been included (36 sessions). Thank you for the comment.

In page 4, line 120. "Its physical fitness and functional capacity was recorded". I recommend explaining better or detailing this information.

  • This sentence has been eliminated, since it is later clarified how the physical condition and functional capacity were evaluated. Thank you for the correction

Trained group
Some information are missing. The session were supervised? the participants training by groups, or alone? if they trained alone, how was the control/record of the exercise program?.

  • The training was conducted by a professional Physical Educator. A paragraph has been included clarifying where the trainings were conducted and in groups of 20 subjects.

Results

Could the authors have more information about the fat mass, lean mass or muscle mass of the participants?.

  • This information has been included. Thank you for the consideration.

In anthropometric results, the authors should be more informative about the changes at the end of the intervention.

  • This information has been included. Thank you for the consideration.

If the gait speed test could be used to diagnosis of sarcopenia. The author estimated the prevalence in this population? at the end of the intervention how many people improve?

  • Thank you for your comment, but the sample of this study showed a gait speed above 0.8 meters / second, considered non-sarcopenia, both before and after the intervention. For this reason, the prevalence of sarcopenia, or change, is not shown.

In page 7, line 234. Homogenize EG / TG

  • It has been revised and replaced. Thank you for the correction

Discussion

In page 8, line 268. A 12 weeks, "low to moderate-intensity exercise program". What intensity was applied?

  • The requested information has been included in the text. Thank you for the suggestion

Thank you very much for the corrections for the improvement of our paper.

Thank you.

Reviewer 2 Report

Thank you very much for allowing me to review this study. The topic worked on is interesting and the effort made by the authors is appreciated.
Once the text has been reviewed, I will now comment on some aspects that could improve its quality.

Line 42: correct
From line 39 to 49, the ideas are repeated. I advise unifying criteria and modifying its wording.
From lines 39 to 68 I see repetition of ideas, as well as disorder in the exhibition, because it returns to mention data that have already been worked on previously (in terms of characteristics of the pathologies to be studied / related to the study). Further specification and reorganization of the introductory section would be requested.

Material and methods.
- Duration of 12 weeks. What is the real time period? (month, year)
- Exclusion criteria. It is recommended to include the following comment: "no regular attendance at the proposed sessions."
- Was the person who carried out the intervention of the experimental group external to the study?
A more detailed explanation of the activities of the experimental group would be appreciated.
How can they verify that the control group, when following their normal routine, does not include light physical activity (such as walking), which can alter the result obtained?

Results:
Be careful with the statements that it makes of "statistically significant results" because for similar values ​​between the control group and the experimental group, in one it says that there were changes, and in the other it says that there were no. Please clarify this fact.

Discussion:
Line 203: says previous studies, when only one study appears in the reference indicated.

References 4, 9, 15, 29, 35, 36, 38, 57, 58, 60 and 63, please review and adapt them to the publication standards.
Reference 68 could update it.

Thank you.

Author Response

REVIEWER 2

Comments and Suggestions for Authors

Thank you very much for allowing me to review this study. The topic worked on is interesting and the effort made by the authors is appreciated.
Once the text has been reviewed, I will now comment on some aspects that could improve its quality.

 Line 42: correct
From line 39 to 49, the ideas are repeated. I advise unifying criteria and modifying its wording.

  • The paragraph has been revised and restructured. Thank you for the suggestion

From lines 39 to 68 I see repetition of ideas, as well as disorder in the exhibition, because it returns to mention data that have already been worked on previously (in terms of characteristics of the pathologies to be studied / related to the study). Further specification and reorganization of the introductory section would be requested.

  • This paragraph has incorporated the information requested by reviewer 1. In addition, it has been revised and restructured in response to your suggestions. Thank you.

Material and methods.
Duration of 12 weeks. What is the real time period? (month, year)

  • The date of the study has been incorporated. Thank you very much for the suggestion

Exclusion criteria. It is recommended to include the following comment: "no regular attendance at the proposed sessions."

  • That phrase has been included, because it was one of the criteria for exclusion, thank you very much for your suggestion.

Was the person who carried out the intervention of the experimental group external to the study?

  • No, the person who conducted the trainings is an expert, member of the research group

A more detailed explanation of the activities of the experimental group would be appreciated.

  • It has been reviewed and modified, also taking into account the suggestions of the first reviewer. Thank you

How can they verify that the control group, when following their normal routine, does not include light physical activity (such as walking), which can alter the result obtained?

  • Your consideration is very interesting. It is something that we take into account because it has not been possible to control in this study. We want to acquire accelerometers and in future studies also control the physical activity of the control group. Thank you very much for your consideration.

Results:
Be careful with the statements that it makes of "statistically significant results" because for similar values ​​between the control group and the experimental group, in one it says that there were changes, and in the other it says that there were no. Please clarify this fact.

  • Thank you for your review. We are checking and we hope that everything is correct.

Discussion:
Line 203: says previous studies, when only one study appears in the reference indicated.

  • It has been reviewed and modified in the text. Thank you for your suggestion

References 4, 9, 15, 29, 35, 36, 38, 57, 58, 60 and 63, please review and adapt them to the publication standards.

  • The references have been reviewed and adapted to the standards of the journal.

Reference 68 could update it.

  • -The reference has been updated. Thanks for the suggestion

Thank you very much for the corrections for the improvement of our paper.

Thank you.

Round 2

Reviewer 1 Report

Some comments for the authors:

Pag 3, line 98: Is it correct the period of study? I think that the year have a mistake, because December 2020 has not happened.

Pag 3, line 100: If the diet was evaluated. What were the most relevant results?

Table 2. I suggest that kg/m2 is enough to abbreviate BMI. Weight[kg])/height [m]2 could be added to the text (Pag 4; lines 136-137). Was the BMI adjusted by kcal consumption? 

Author Response

REVIEWER 1 – Review 2

Comments and Suggestions for Authors

Some comments for the authors:

Pag 3, line 98: Is it correct the period of study? I think that the year have a mistake, because December 2020 has not happened.

  • We apologize to the reviewer for the numerical error of the year. It has already been corrected in the document. Thank you very much for your correction

Pag 3, line 100: If the diet was evaluated. What were the most relevant results?

  • We apologize, this sentence has been change.

Table 2. I suggest that kg/m2 is enough to abbreviate BMI. Weight[kg])/height [m]2 could be added to the text (Pag 4; lines 136-137).

- It has been corrected in the text and in the table. Thank you very much for your review

Was the BMI adjusted by kcal consumption? 

  • Thank you for your considerations. It will be taking into account in futures researchers.

On behalf of the authors we are very grateful to the reviewer for his corrections, as they have helped to improve the quality of the manuscript

Reviewer 2 Report

I appreciate your corrections. once the new manuscript has been reviewed, I have verified that the suggested modifications have been made.

Author Response

REVIEWER 2 Review 2

I appreciate your corrections. once the new manuscript has been reviewed, I have verified that the suggested modifications have been made.

Thank you very much.
